# Epigenetic Regulation of TRAIL Signaling: Implication for Cancer Therapy

**DOI:** 10.3390/cancers11060850

**Published:** 2019-06-19

**Authors:** Mohammed I. Y. Elmallah, Olivier Micheau

**Affiliations:** 1INSERM, Université Bourgogne Franche-Comté, LNC UMR1231, F-21079 Dijon, France; 2Chemistry Department, Faculty of Science, Helwan University, Ain Helwan 11795 Cairo, Egypt

**Keywords:** histone deacetylase (HDAC), histone deacetylase inhibitors (HDACIs), chromatin remodeling, cancer, tumor necrosis factor (TNF), TRAIL, methylation, silencing

## Abstract

One of the main characteristics of carcinogenesis relies on genetic alterations in DNA and epigenetic changes in histone and non-histone proteins. At the chromatin level, gene expression is tightly controlled by DNA methyl transferases, histone acetyltransferases (HATs), histone deacetylases (HDACs), and acetyl-binding proteins. In particular, the expression level and function of several tumor suppressor genes, or oncogenes such as c-Myc, p53 or TRAIL, have been found to be regulated by acetylation. For example, HATs are a group of enzymes, which are responsible for the acetylation of histone proteins, resulting in chromatin relaxation and transcriptional activation, whereas HDACs by deacetylating histones lead to chromatin compaction and the subsequent transcriptional repression of tumor suppressor genes. Direct acetylation of suppressor genes or oncogenes can affect their stability or function. Histone deacetylase inhibitors (HDACi) have thus been developed as a promising therapeutic target in oncology. While these inhibitors display anticancer properties in preclinical models, and despite the fact that some of them have been approved by the FDA, HDACi still have limited therapeutic efficacy in clinical terms. Nonetheless, combined with a wide range of structurally and functionally diverse chemical compounds or immune therapies, HDACi have been reported to work in synergy to induce tumor regression. In this review, the role of HDACs in cancer etiology and recent advances in the development of HDACi will be presented and put into perspective as potential drugs synergizing with TRAIL’s pro-apoptotic potential.

## 1. Introduction

Cancer is considered as the leading cause of death worldwide. It has been reported to occur as a result of epigenetic modifications, including amplifications, translocations, deletions, and point mutations [1,2]. These epigenetic modifications are linked to abnormal cellular transformation, characterized, among other things, by uncontrolled proliferation and resistance to cell death, forming a lump or what is called a tumor. In addition, DNA methylation and the post-translational modification of proteins such as histones, including acetylation and methylation, are also believed to play a central role in tumorigenesis by modifying the structure of chromatin and the subsequent negative regulation of tumor suppressor genes or oncogenes without any change in the DNA sequence [3]. The main function of histones is the packaging of genomic DNA inside the nucleus. Histone proteins are rich with positively charged amino acids, lysine and arginine, which makes their overall structure positive. In this context, histones can interact with the negatively charged phosphate group of DNA. These proteins are composed of several types of histone including H1, H2A, H2B, H3, and H4. H2A, H2B, H3, and H4 represent the main histone core, while H1 is known as a linker histone [4]. The core multimeric protein is an octamer molecule, mainly consisting of two copies of each histone type (2H2A, 2H2B, 2H3, and 2H4) forming a globular structure called the nucleosome (Figure 1a). Each nucleosome is wrapped by approx. 146 bp of DNA and separated by 50 base pair (bp) linker DNA. The N-terminal domain of histones possesses an unstructured part that has also been suggested to be involved in the epigenetic modification of chromatin. Considering the process by which histone proteins are post-translationally modified, chromatin adopts various structural conformations that regulate both the repression and activation of gene transcription (Figure 1b and [5]).

Acetylation/deacetylation of the N-ε-lysine residues of histones is regulated by a group of acetylating and deacetylating enzymes. Histone acetyltransferases (HATs) transfer an acetyl group from an acetyl-coA molecule to the N-ε-lysine residues of histone, resulting in the neutralization of its positive charge and in the activation of gene transcription. In contrast, histone deacetylases (HDACs) remove the acetyl group from the N-ε-lysine residues of histone providing a tight interaction between DNA and histone protein (Figure 1b). In this case, the chromatin exhibits more compacted conformation [6]. Excessive histone deacetylation induced by HDAC can have a significant impact on the pathology of cancer via the silencing of tumor suppressor genes, including p53. For instance, the more the chromatin is condensed, the fewer promoters of these target genes are accessible to transcription factors, whereas when cells exhibit higher HAT activity, chromatin relaxation is associated with an increased transcriptional activation of tumor suppressor genes [7,8]. Given that dysregulation of HDAC and/or HAT function has been associated to cancer etiology, intensive research is being conducted worldwide to develop HDAC inhibitors. In this review the role of HDAC and epigenetic modifications in cancer will be discussed with a special emphasis on the potential interest of HDAC inhibitors for immunotherapies based on TRAIL or TRAIL-derivatives.

## 2. Classification and Localization of HDACs

HDAC can be classified into four groups according to their homology with yeast function and sequence (Figure 2). Class I histone deacetylases contains four proteins ranging from 42 to 45 kDa; known respectively as HDAC1, 2, 3 and 8, which are mostly localized in the nucleus. Class II HDACs are larger in size (120-130 kDa) and includes HDAC4, 5, 6, 7, 9, and 10. Class II histone deacetylases have been further subdivided in two groups, Class 2a that encompasses HDAC4, 5, 7, and 9, which display high sequence homology. HDAC6 and 10 have been grouped in class 2b due to the presence of an extra catalytic domain. Class II HDAC members are mainly localized in the cytoplasm, however they shuttle from the cytoplasm to the nucleus when phosphorylated. Class III histone deacetylases are also known as sirtuins due to their homology with the yeast silent information regulator 2 (Sir2) gene and to their requirement of NAD^+^ for their enzymatic activity. Class IV is considered as a category in its own, with HDAC-11 as a single member. Similar to class I and II members, HDAC11 is a metalloenzyme that contains Zn^+2^ ion in its binding pocket [9,10,11,12,13,14]. 

Besides histones, HDACs can also deacetylate non-histone proteins, some of which are transcription factors such as E2F1, c-Jun, GATA1, TFIIE, TFIIF, p65 (RelA), transcription repressors like Yin Yang 1, Mad/Max, the tumor suppressor gene p53 [12,15,16], and the oncogene c-Myc [17] to cite a few. Other non-histone proteins are frequently regulated by HDACs such as the steroid hormone receptor binding protein, the cytoskeletal protein (α-tubulin), nuclear transport (importin-α7), DNA helicase (WRN), signal transduction (β-catenin), and the heat shock protein (Hsp90) [18,19,20,21]. On the other hand, like deacetylation, hyperacetylation of given proteins can change their function. Likewise, hyperacetylation of p53 was shown to increase its stability and subsequent induction of apoptotic cell death [18,20,21,22]. Moreover, acetylation of Hsp90 leads to the activation of the proteosomal degradation of the receptor tyrosine-protein kinase (ErbB2) oncoprotein [23]. Non-histone protein acetylation can also be induced by a large majority of histone acetyl transferases (HATs). Thus, HATs have recently been renamed as lysine acetyl transferases (KATs). This family of enzyme has been divided into two main groups (Figure 3). Type A is the larger group and includes several sub-families. The members of type A HATs are mostly localized in the nucleus and classified into families according to their sequence homology. The Gcn5-related N-acetyltransferases (GNAT) family, includes both KAT2A and KAT2B. The p300/CBP family consists of KAT3A and 3B. 

The MYST family represents the largest HAT family, which includes KAT5, KAT6A, KAT6B, KAT7, and KAT8. Type B HATs are abundant in the cytoplasm such as KAT1 and KAT4. The members of both CBP/P300 and GNAT families contain a bromodomain (BD) that mainly binds to the acetylated lysine rich region of histone proteins. Moreover, the members of CBP/P300 and MYST family members possess a cysteine rich, zinc-binding domain which facilitates binding to acetyl group. Some MYST family HATs contain in addition an N-terminal chromodomain, which binds to methylated lysine residues [24]. HATs can also acetylate non-histone proteins that could influence their promotor activities and specificities, such as β-catenin, Myc proto-oncogene protein (C-MYC), tumor suppressor protein p53, and the nuclear factor kappa-light-chain-enhancer of activated B cells (NF-κB) [25,26,27].

## 3. Molecular Mechanism of HDAC Action

HDACs are involved in the epigenetic modification of histone proteins via removal of acetyl group of N-ε-lysine residues of histone. Considering that HDACs induce hypoacetylation, the distance between the nucleosome molecule and chromatin DNA is markedly decreased and hence the histone adopts a compacted structure of wrapped DNA (Figure 1). Chromatin compaction impedes the access of transcription factors to the promoter of the target genes present within the area, including tumor suppressor genes, resulting in uncontrolled cell proliferation and cancer development. The catalytic domain of HDAC consists of approx. 390 amino acid residues (Figure 2), which comprises a set of conserved amino acids. The substrate binding site of HDAC adopts a curved tunnel with a wider bottom, similar to Zn^+2^-binding proteins like thermolysin and carboxypeptidase [28,29,30]. The structural environment of the HDAC substrate binding site encompasses two adjacent histidine residues (His142 & His 143), two aspartic acid residues (Asp176 & Asp183), and one tyrosine residue (Tyr306). Deacetylation process occurs through a charge-relay system in the presence of Zn^+2^ ions. The carbonyl group of the acetyl lysine residues is polarized for nucleophilic attack as a result of coordination with Zn^+2^ bonded water molecules and hydrogen bonding with Tyr306 [31,32]. Zn^+2^ coordinated residues along with substrate binding constitute a tetrahedral structure, mainly consisting of five coordinates including water molecules. This step was reported to be the rate limiting step in the hydrolysis of the acetyl group of lysine residue by HDAC8 [30]. Furthermore, activation of the nucleophile is mediated by the general base character of the adjacent His142 and His134 residues in the substrate binding site, which are influenced by hydrogen bonding with Asp176 and Asp183, respectively [33]. H143 residue is localized in a close vicinity to the leaving amino group of acetyl lysine. Therefore, it acts as a proton donor and hence mediates the separation of the amino group as a result of tetrahedral structural conformation [34].

## 4. Physiological Function of HDAC

HDACs are recruited to the gene promoter in the form of multiprotein complex (see examples Figure 4 and [35]). The interacting protein partners of HDACs are mostly acting as transcriptional coregulators. They include proteins such as mSin3 interacting domain of the paired amphipathic helix protein (sin3a), the nuclear receptor co-repressor 1 (N-CoR), and the nuclear receptor co-repressor 2/ silencing mediator for retinoid and thyroid hormone receptors (SMRT). The binding of a specific DNA sequence by transcriptional regulatory proteins along with HDACs can also occur with the help of epigenetic modifying proteins such as methyl binding proteins (MBDs), DNA methyl transferases (DNMTs), and histone methyl transferases (HMTs). For example, the MBDs, methyl-CpG binding domain (MeCP2), is believed to recruit HDAC to methylated promoters resulting in transcriptional repression (Figure 4a and [36]). The receptor interacting protein 140 (RIP140) is a negative co-regulator, which binding to HDAC1 and HDAC3 represses expression of nuclear receptors (Figure 4b and [37]). Hyperacetylation of histones can also occur independently of the loss of HDAC activity. For example, knockout of DNMT1 revealed a significant increase of the acetylated histone H3 and a decrease in the level of the methylated form. The decreased level of methylated H3 was mainly attributed to the dissociation of HDAC-DMNT1 complex leading to hyperacetylation of H3 [38].

Moreover, HDACs were reported to regulate neuronal differentiation via direct interaction with the N-terminal domain of DMNT3b. Inhibition of HDACs in pheochromocytoma cell line (PC12) prevented cell differentiation stimulated by nerve growth factor (Figure 4b). The case was reversed when DMNT3b was overexpressed in the same cell line in which a significant cell differentiation was detected [39]. HDAC can also interact with transcription factors and nuclear receptors other than MBDs. The interaction of HDAC with the cell cycle regulator retinoblastoma protein (Rb) leads to uncontrolled cell progression [40]. Rb is a family of tumor suppressor proteins involved in the inhibition of cell growth via the suppression of cell cycle progression. When a cell is ready to divide, Rb is inactivated by phosphorylation allowing cell cycle progression. Defects in Rb proteins were found to play a central role in the initiation of cancer [41]. Transcription factor E2F belongs to a protein family that is responsible for cell cycle control and the synthesis of DNA in eukaryotes. It consists of nine members, of which three (E2F1, E2F2, and E2F3a) are transcription activators and the remaining six (E2F3b, E2F4, E2F5, E2F6, E2F7, and E2F8) are cell cycle repressors. The gene promoters of E2F transcription repressors was suggested to be inactivated by HDAC-Rb complex in transformed cells [39,42,43]. It has been reported that the interaction between HDAC and Rb promotes uncontrolled cellular proliferation and the initiation of cancer. Trichostatin A (TSA), a classical HDAC inhibitor (HDACI) is able to prevent HDAC-Rb complex assembly hence repressing the transcription of genes essential for cell cycle progression [44]. 

In addition, HDAC has the ability to bind nuclear receptors, thus mediating the regulation of gene expression. For example, estrogen receptors (ERs) are nuclear receptors involved in the transcriptional regulation of genes that determine the function of both breast and ovary. For instance, the nuclear receptor ER-α is activated by its ligand 17β-estradiol, which is implicated in the transcriptional regulation of estrogen responsive genes in different cell types. Such an interaction was found to mediate the normal development of breasts as well as progression of breast cancer. The downregulation of ER-α expression in the breast epithelium is critical for the initiation of breast cancer. The reason behind was mainly assigned to the in vivo interaction between HDAC1 and ER-α in its promoter region suppressing its transcriptional activity (Figure 4b). This interaction can be reversed by the availability of estrogens and also in the breast cancer MCF7 cell line with the HDAC inhibitor TSA, resulting in the upregulation of ER-α on both mRNA and protein levels [45]. Macaluso and coworkers [46] demonstrated that the transcriptional silencing of ER-α is mediated by two multiprotein repression complexes (1) pRb2/p130-E2F4/5-HDAC1-histone methyl transferase (SUV39H1)-p300 and (2) pRb2/p130-E2F4/5-HDAC1-DNMT1-SUV-39H1 that are recruited to the ER-α promoter of breast cancer cells.

## 5. Role of HDAC in Cancer

Strong evidence suggests that defects in the DNA methylation as well as defects in the post-translational modification of histones are directly causative of cancer initiation in humans. Indeed, the loss of acetylated lysine residue (Lys16) and trimethylated Lys20 of histone H4 is considered as an initial event for the development of cancer [47]. Furthermore, the decreased rate of histone acetylation in gastrointestinal tumors was also found to be associated with invasion and metastasis [48]. The mechanism by which HDACs deacetylate lysine residues of histone proteins still remains enigmatic. However, hypoacetylation due to decreased levels of HATs was mainly attributed to either mutations or chromosomal translocation such as leukemias. This was found to be accompanied by increasing in HDAC activity [49]. The most available data regarding the mechanism of HDAC in cancer is restricted in the formation of HDAC-fusion protein complex during binding to a specific gene promoter. These fusion proteins are considered as an end product of chromosomal translocation in case of hematological malignancies [50]. Acute promyelocytic leukemia (APL) is a subtype of acute myelocytic leukemia (AML), known as the cancer of white blood cells. The genotypic character of APL is represented in the chromosomal translocation between chromosome 15 and 17 including the retinoic receptor-α (RARα). Some fusion proteins like promyelocytic leukemia/retinoic acid receptor (RAR-PML) and retinoic acid receptor promyelocytic leukemia zinc finger protein (RAR-PLZF) are recruited along with HDAC to a specific DNA sequence located in the promoter region (Figure 5) known as retinoic acid-responsive elements (RAREs). Consequently, the expression of genes that are responsible for the normal differentiation and proliferation of myeloid cells is inhibited [51,52]. 

Several studies have reported the overexpression of HDAC in different types of cancer such as HDAC1 in gastric [53], prostate [54], colon [55], and breast [56] cell carcinomas. Overexpression of HDAC2 was also reported in cervical [57], gastric [58], and colorectal cancers [59]. HDAC3 and HDAC6 were found to be overexpressed in colon and breast cancers [54,60]. In this context, overexpression of HDAC can lead to the transcriptional repression of tumor suppressor genes via recruitment of HDAC in the form of multiprotein complex to their specific promoter regions. Cyclin-dependent kinase inhibitor (p21^WAF1^) is a cell cycle progression inhibitor. Expression of p21^WAF1^ is significantly influenced by promoter hypoacetylation. Inhibition of HDAC significantly increases the acetylation of p21^WAF1^ promoter and subsequent inhibition of tumor growth [61].

Expression of p21 is also influenced by p53 that competes with HDAC1 to specific promoter region of *p21* known as promoter-specific RNA polymerase II transcription factor (Sp1). This results in the release of HDAC1 from Sp1, leading to an increase in the expression of p21 ([62] and Figure 6). Inhibition of HDAC increases the acetylation of p53 and enhances its stability, which subsequently strengthens the interaction with *p21* promoter. Moreover, HDAC inhibitors have been described to induce hyperacetylation of sp1 [63] and sp3 [64] and to change the promoter acetylation profile and expression levels of several death receptors involved in the transduction of apoptotic signals (Figure 6). These include the tumor necrosis factor-related apoptosis-inducing ligand (TRAIL) death receptors (DRs) DR4 [65,66,67] and DR5 [66,68,69,70,71], as well as FAS ligand/CD95 ligand (FASL) [72,73], and FAS [74]. 

Class III HDAC/sirtuins are of growing interest in oncology due to their ability to regulate gene expression, apoptosis, stress responses, genome integrity, and cancer metabolism [75,76,77,78,79,80,81,82,83,84]. Lys16 residue of H4 (lys16-H4) and Lys9-H3 were reported to be the substrate of this group of HDACs [85,86]. Dysregulation of their expression levels has been described in cancer cells associated or not with oncogenic or tumor suppressor functions. For example, SIRT1 was found to be highly expressed in human lung cancer [87], prostate cancer [88], and leukemia [89], however its expression is reduced in colon cancer, when compared to normal tissues [90]. Mechanistically, sirtuins such as SIRT1 are able to deacetylate p53 leading to the inhibition of its DNA damage functions [91] or to induce the hypoacetylation of the DNA repair enzyme Ku70, enhancing its non-homologous end joining DNA repair ability [92,93] and allowing the survival of cancer cells [21]. On the contrary, treatment with SIRT1 inhibitors leads to the increased expression of tumor suppressor genes and increased level of the acetylated lys16-H4 and lys9-H3 in both colon and breast cancer cell lines [86].

Inhibition of HDAC gained the attention of several research groups in the field of cancer drug discovery, making HDAC a promising drug target for the treatment of cancer [94,95]. HDACIs are suggested to induce apoptosis by inhibiting multiple signaling pathways. The effect of HDACIs is not restricted to histone proteins, these inhibitors can also directly impact non-histone proteins [48,96]. They are categorized into two categories: (1) HDAC isoform-selective inhibitors, which target several types of HDAC, and (2) pan-inhibitors, which act against all type of HDACs [97]. Clinical trials have been conducted for various HDAC inhibitors against different type of tumors. These inhibitors are divided into four different classes based on their chemical structures (Figure 7), which include (I) hydroxamic acids, (II) short chain fatty acids, (III) benzamides, and (IV) cyclic peptides [98]. The group of hydroxamic acid-based HDACIs assessed in clinical trials include abexinostat, belinostat, givinostat, pracinostat, panobinostat, quisinostat, resminostat, and vorinostat [99]. Trichostatin A (TSA) and suberoyl bis hudroxamic acid also belong to the hydroxamic acid group. 

TSA is a natural HDAC inhibitor that was originally isolated from actinomycete *Streptomyces hygroscopicus* and is used for its antifugal property. Like belinostat, panobinostat, vorinostat and TSA are able to inhibit all classes of HDACs. However, TSA is solely used in laboratories due to the fact that this pan-HDAC inhibitor displays high toxicity [100]. Valproic acid (VPA), butyric acid and phenylbutyric acid, belong to the HDACi class of short-chain fatty acids (Figure 7). VPA targets class I and IIa HDACs [101], whereas butyric and phenylbutyric acids target HDACs of class I and II [96]. This class of HDAC inhibitors is probably the least potent of the family of HDAC inhibitors. Benzamides, represent another subclass. It includes the class I HDACi’s entinostat, tacedinaline, and 4SC202, as well as mocetinostat a class I and IV HDAC selective inhibitor. The last subgroup of HDAC inhibitor is represented by romidepsin, a class I HDAC inhibitor. 

To date, only four compounds have been investigated as HDAC inhibitor and approved by US Food and Drug Administration (FDA) in clinical trials. These compounds, including vorinostat (suberoylanilide hydroxamic acid, SAHA) and Romidepsin (cyclic tetrapeptide) which revealed a promising HDAC inhibition against different type of cancers were approved for the treatment of cutaneous T-cell lymphoma [102,103,104,105]. Belinostate (Class I) was approved for the treatment of peripheral T-cell lymphoma [106]. Panabinostat is a member of the hydroxamic acid group and has recently been tested in clinical trials and approved for the treatment of multiple myeloma [96]. Other inhibitors are under development.

Class III HDAC inhibitors, which selectively target the NAD^+^-containing HDACs sirtuins have mostly been implemented in the treatment of various clinical diseases like cardiovascular disorders, aging, neurodegenerative disorders, and cancer [107]; however, recent studies have reported the investigation of sirtuins, particularly sirtinol, cambinol, and EX-527 as therapeutic agents for cancer [108,109]. It should be noted though that cambinol was found to display stronger affinity to neutral sphingomyelinase-2 than class III HDACs [110], questioning its specificity for sirtuins. 

The anticancer effect of HDAC inhibitors have been found to involve a number of molecular events including inhibition of angiogenesis, generation of reactive oxygen species (ROS), and apoptosis (Figure 8). Inhibition of angiogenesis by HDAC inhibitors was found to prevent cancer metastasis. Their anti-angiogenic properties were associated with their ability to inhibit pro-angiogenic proteins such as vascular endothelial growth factor (VEGF), endothelial nitric oxide synthase (eNOS), bFGF and TGFbeta1 [111,112,113], to cite a few.

Some inhibitors such as entinostat, also known as MS-275, were also found to induce radical oxygen species (ROS) leading to cell death in leukemia cells through the regulation of BH3 interacting domain death agonist protein (Bid) and B-cell lymphoma-2 (Bcl-2) expression levels [114,115]. Mechanistically it was found in prostate (LNCaP), breast (MCF7), and bladder (T24) cancer cell lines that both vorinostat and entinostat induce ROS by inhibiting the binding protein-2 (TBP-2), an inhibitor of the cellular thioredoxin (Trx) [116]. Interestingly, the induction of ROS in leukemia cells by vorinostat was also associated with activation of caspase-10 and Hsp90 cleavage [117]. 

More specifically, cellular cell death induced by HDAC inhibitors is largely associated with their ability to regulate selectively both the intrinsic and extrinsic pro-apoptotic pathways in tumor cells [118,119,120,121], but not in normal cells [122,123]. Vorinostat for example is able to induce the transcriptional activation of pro-apoptotic genes like Bcl-2-associated X protein (Bax), Bcl-2 associated agonist of cell death protein (Bad) and Phorbol-12-myristate-13-acetate-induced protein 1 (Noxa), promoting apoptosis through mitochondrial membrane changes and leading to the release of the pro-apoptotic factors such as Diablo/second mitochondria-derived activator of caspases (Smac), ATP and Apoptotic protease activating factor 1 (Apaf1), caspase-9, Serine protease (HtrA2), and cytochrome c (cyt c) in transformed fibroblast cell ([123] and Figure 8). 

Other BH3-containing pro-apoptotic Bcl2 family members have been described to account for histone deacetylase inhibitor-induced cell death, including Bmf [124], Bak [125], Bid [126], and PUMA [125,127]. Along the line, it was found that Apaf1 expression level is significantly upregulated by the HDAC inhibitor TSA or siRNAs targeting HDAC1, HDAC2 and HDAC3 in hepatocellular carcinoma cell lines [128]. Concomitantly, HDAC inhibitors are also able to trigger apoptosis by down-regulating the anti-apoptotic proteins of the Bcl-2 family such as Bcl-2 itself [129], Bcl-xL [130,131], Bcl-xS [132], Mcl-1 [115,131], as well as XIAP [115,131] and survivin [133,134], two cellular inhibitors acting downstream of the mitochondria. Remarkably, the tumor properties of histone deacetylase inhibitors, consistent with their selective effect on pro-apoptotic genes involved in the intrinsic cell death pathway, also regulate genes associated with the extrinsic pathway, largely regulated by receptors of the TNF superfamily, including FasL, TRAIL and their cognate receptors ([132,135,136,137] and Figure 8).

## 6. DNA Methylation and Cancer

DNA methylation is another important epigenetic modification that occurs in the mammalian genome leading to regulation of gene expression. It acts in concert with other epigenetic mechanisms, including acetylation, to regulate normal gene expression and is often found to be exacerbated in cancer. Likewise, genome-wide hypermethylation has been described in primary and metastatic tumors [138,139,140,141,142]. Methylation occurs at specific DNA sequences such CpG dinucleotides, representing 5–10% of the human genome [143,144,145]. CpG dinucleotide-rich genomic DNA regions are also known as CpG islands, they are repeated in the human genome every 100 kb. DNA methylation is controlled by a group of DNA methyl transferases (DNMTs) [146] and demethylases (See the following reviews for detail [147,148,149,150,151]) [152]. Dysregulation of TRAIL signal transduction by methylation will be further described in the Section 7.1.

## 7. Epigenetic Regulation of TRAIL Proapoptotic Signaling

TRAIL belongs to the tumor necrosis factor (TNF) family, which encompasses 19 ligands and 29 receptors. These members were found to play a major role in the induction of apoptotic cell death and cell survival as well [153,154,155,156,157]. TNF-α, tumor necrosis factor related apoptosis-inducing ligand (TRAIL), and Fas ligand (FasL/CD95L) are considered as the major cytokines/proteins in this family with proapoptotic potential. The apoptotic pathway mediated by these cytokines occurs through two different signaling pathways, mainly the extrinsic (cytoplasmic) pathway and the intrinsic (mitochondrial) pathway (Figure 9). The extrinsic pathway is initiated when ligands such as FasL and TRAIL bind to their agonistic cell surface cognate death receptors (DR). Consequently, DR tends to oligomerize and induces the recruitment of Fas-associated death domain (FADD) and procaspase-8/-10 to form the death-inducing signaling complex (DISC). Following the assembly of DISC, a cascade of controlled programmed suicide signals is mediated by the activation of cellular caspases including the initiator caspases-8/-10 and the executioner caspases-3, -6, -7. The intrinsic pathway occurs via the cleavage of Bid by caspase-8 into truncated Bid (tBid) which subsequently translocates to the outer mitochondrial membrane in combination with Bax. This results in changing the equilibrium potential of the outer mitochondrial membrane and release of cyt c. Release of cyt c in the cytosol enables its interaction with Apaf1, leading to the formation of a large protein complex known as apoptosome, which recruits and activates the initiator caspase-9, allowing activation of the effector caspase-3, the apoptosis executioner [158,159,160]. In some instances, such as during ER stress, some receptors of this family, including DR4, DR5 and TNFR1 are able to engage apoptosis or necroptosis in a ligand-independent manner by inducing the recruitment and activation of the caspase-8 [161,162,163,164] or through RIPK1/RIPK3 and MLKL [164]. 

Like HDAC inhibitors, TRAIL was found to display selective antitumoral properties [165] and whereas it has raised major interest for cancer therapy [166], it soon became clear that a number of tumor cell lines or primary cells derived from patients suffering from cancer exhibit inherited secondary resistance toward TRAIL-induced cell death [167]. Such resistance may arise due to (1) the downregulation of cell surface DRs, (2) insufficient expression of caspase-8, and (3) overexpression of some antiapoptotic agents including caspase-8 and FADD-like apoptosis regulator (c-FLIP), Bcl-2, B-cell lymphoma-extra-large (Bcl-xL), myeloid cell leukemia 1 (Mcl-1), survivin, and (XIAP), X-linked inhibitor of apoptosis [160]. Epigenetic drugs such as HDAC inhibitors and demethylating agents have been found, however, to regulate the expression levels of both the receptors and downstream signal executioners or regulators of TRAIL (Figure 9). Moreover, histone deacetylase inhibitors have also been found, in some instances, to trigger selective apoptosis in cancer cells through death receptors ([135,137,168,169] and Figure 9). The molecular mechanisms by which these compounds trigger or restore the sensitivity of cancer cells toward apoptosis induced by the extrinsic or intrinsic pathway are often specific of the cell type, the drug itself, and the dose. In the following paragraph we will summarize the current understanding of the molecular mechanisms induced by epigenetic modulators associated specifically with TRAIL, its cognate receptors and downstream proximal signal transduction partners or regulators.

### 7.1. Restoration of TRAIL-Induced Apoptosis by Demethylating Agents 

#### 7.1.1. Regulation of Survivin and XIAP by Methylation

Survivin is one of the most distal inhibitors of TRAIL pro-apoptotic signaling. Its promoter is often hypomethylated in tumors, including hepatocellular, cervical or ovarian carcinomas and astrocytomas or multiform gliomas ([170,171,172,173,174] and Figure 10a). Although, none of these reports investigated the corresponding protein expression levels in these biopsies, contrary to normal counterpart cells, these findings suggest that survivin may be highly expressed in these tumors and thus likely to inhibit TRAIL-induced cell death. Indeed, overexpression of survivin in tumor cells and biopsies, at the protein level, has largely been described in the literature and is most of the time associated with poor outcome, prompting interest in its targeting [175,176]. However, the hypomethylation of the promoter of survivin was also found to allow p53 binding, thus leading to the repression of survivin transcription and protein expression ([177] and Figure 10a). Supporting these findings, the same authors demonstrated that hypermethylation of survivin promoter, in endometrial tumors, was associated with high expression levels of the protein [177]. Demethylating drugs such as decitabine have been shown to be poor inhibitors of survivin expression in small cell lung carcinoma (SCLC) cells [178]. However, the combination of class I & II HDAC inhibitor with decitabine inhibits survivin expression in these cells and increases their sensitivity to TRAIL-induced cell death [178]. The second most important and downstream inhibitor of TRAIL, XIAP, is not directly regulated by methylation, but its regulator, XIAP-associated factor 1 (XAF1), which is itself negatively regulated by hypermethylation [179]. As with survivin, the methylation status of XAF-1 promoter is not the only determinant of its basal transcription expression level, since it was found that restoration of XAF1 expression could be achieved by stimulating cells with IFN-beta, in spite of sustained promoter hypermethylation in these cells [180]. 

#### 7.1.2. Regulation of Initiator Caspases by Methylation

Further upstream are the caspase-8 and caspase-10, two cysteine proteases whose expression is required to enable TRAIL-induced apoptosis. Deregulation of the caspase-8 expression associated with *CASP8* gene promoter hypermethylation has been described in neuroblastoma [181], retinoblastoma (Rb) [182], von Hippel-Lindau, and sporadic phaeochromocytoma cells [183]. Contrary to survivin or even XAF1, treatment of corresponding cells with decitabine, alone, restores both expression levels of caspase-8 and sensitivity to TRAIL-Induced apoptosis [181,182,184,185]. Methylation-induced regulation of caspase-8 expression has also been described in SCLC [178,186] and hepatocellular carcinomas (HCC) [187,188]. Of interest, silencing both DNMT1 and DNMT3b in HCC was found to be sufficient to restore caspase-8 expression and sensitivity to TRAIL-induced cell death ([188] and Figure 10b). In another study performed on 76 patients associated with glioblastomas, the loss of caspase-8 by methylation of its promoter reached more than 50% of the patients [189].

#### 7.1.3. Regulation of TRAIL Receptors by Methylation

Besides caspase-8, the most important genes regulated by methylation and impacting TRAIL-induced cell death are those encoding its cognate receptors, namely DR4, DR5, DcR1 and DcR2. One of the earliest study performed on pediatric neuroblastoma cell lines, represented by nine neuroblastomas, three peripheral primitive neuro-ectodermal tumors (PNETs), and three cell lines from adult tumors, demonstrated that a loss of *DcR1*, and *DcR2* mRNA expression occurred in 86% and 66% of tumor cells respectively, due to methylation of their promoter (Figure 10b). Methylation of *DR4* and *DR5* was also demonstrated, but to a much lesser extent, 47% and 20%, respectively [190]. Inhibiting methylation using 5-Azacytidine (Vidaza), restored in all cases transcription of *DcR1* and *DcR2* in these cells. Remarkably, methylation of *DcR1* and *DcR2* promoters was not observed in normal tissues, originating from heart, liver, lung, muscle, ovary, spleen or kidney [190,191]. 

Aberrant methylation of *DR4* (41%), *DR5* (10%), *DcR1* (24%), and *DcR2* (26%) were also described in von Hippel-Lindau and sporadic phaeochromocytomas [183]. The same authors also found out that similar methylation profiles could be detected in neuroblastomas and that 5-aza-2′-deoxycytidine (decitabine), by inhibiting promoter methylation of these genes could restore their expression. In glioblastomas, like *CASP8,* more than 40% of the cell lines studied displayed significant methylation in *DR4* gene promoter, explaining at least partially the heterogeneity of the disease [189]. 

A high throughput quantitative DNA methylation analysis, performed on 17 gene promoter regions associated with DNA damage response (DDR) and death receptor apoptotic pathway with 162 normal and cancerous breast tissues from 81 sporadic breast cancer patients, demonstrated that sporadic breast tumor tissues displayed an obvious hypermethylation of TRAIL receptor genes *DR5*, *DcR1*, *DcR2* associated with an hypomethylation of *DR4* [192]. In another study focusing on invasive cervical cancers (9 cell lines and 114 primary tumors), it was demonstrated that *DcR1* and, to a lesser extent, *DcR2* are hypermethylated, but not *DR4* and *DR5* [193]. Hypermethylation of *DcR1* gene promoter was found to be associated with 56 (45.5%) of the 123 invasive cervical cancers (CC). Methylation in *SCLC* was less frequently detected with 5.7% of the investigated tumors [193]. While it remains unclear why these receptors are differentially regulated by methylation in different tumor cell types and whether this has any influence toward TRAIL-induced apoptosis, a number of studies provide evidence that indeed silencing of TRAIL agonist receptors can impair TRAIL-induced cell death. It has been demonstrated for instance that silencing DR4 by methylation in astrocytic gliomas impedes TRAIL-induced cell death [194]. The resistance of cancer cell lines to TRAIL attributed to the hypermethylation of the promoter region of *DR4* was abrogated by decitabine and associated with the re-expression of DR4 at the cell surface [194]. Methylation of *DcR1* promoter was demonstrated in malignant melanoma cells to require the DNA methyl transferase DNMT1, whereas *DcR2* methylation involved both DNMT1 and DNMT3a ([195] and Figure 10b). In breast cancer cells, however *DcR2* methylation was proposed to be solely mediated by DNMT1 [196]. Like *DcR2*, methylation of *DR5* may be governed by DNMT1 and DNMT3a [188,197]. Transcriptional regulation of *DR4* and *DR5* has, in addition been demonstrated to be also tightly controlled by two histone lysine demethylases, namely KDM2B [198] and KDM4A ([199], respectively (Figure 10b). Differential activity or expression of these DNA and histone methyl transferases is thus likely to account for selective expression of TRAIL receptors in tumor cells. Likewise, the DNMT1 inhibitor, brominated alkaloid Isofistularin-3 (Iso-3), from the marine sponge Aplysina aerophoba restores selectively DR5 but not DR4 expression in the B lymphoma cell line Raji, increasing its sensitivity to TRAIL-induced cell death [197]. In the same vein, silencing DNMT1 and DNMT3b in human hepatoma cells restores DR5 expression and TRAIL sensitivity [188]. 

Inhibition or silencing of KDM4A or KDM2B was also found to be sufficient to restore DR4 and DR5 expression, respectively and enhance or confer sensitivity to TRAIL [198,199]. It should nonetheless be mentioned here, that in addition to DR4 or DR5, the effects associated with the regulation of the methylation status of their promoter or histone neighborhood can also involve the coordinated negative regulation of antiapoptotic proteins such as c-FLIP, survivin, XAF1 or Mcl-1 [198], or the upregulation of pro-apoptotic proteins such as TRAIL itself [199]. 

### 7.2. HDACIs Sensitize Tumor Cells to TRAIL-Mediated Apoptosis

#### Regulation of Gene Expression by HDACi

As illustrated in Figure 9 and Figure 11c, HDACi have often been described to enhance apoptosis induced by death-domain-containing receptors of the TNF family by coordinating proapoptotic and anti-apoptotic gene expression. For the sake of clarity, only the effects of HDACi on TRAIL-induced cell death will be presented here. 

In melanoma cell lines, the HDAC 1 and 3 inhibitor suberoyl bis-hydroxamic acid (SBHA) is able, alone, to enhance TRAIL-induced cell death by coordinating the upregulation of the pro-apoptotic proteins caspase-8, caspase-3, Bid, Bak, Bax, and Bim, while downregulating at the same time the antiapoptotic proteins, Bcl-xL, Mcl-1, and XIAP [126,131,200]. VPA, SAHA, and TSA on several melanoma cell lines were found to allow acetylation of H3 and H4 histones, leading to an increase in DR4 and DR5 expression levels and to concomitant inhibition of Bcl-xS, Bcl-xL [132]. Keeping in mind that immunotherapy has dramatically changed the treatment paradigm of melanoma, it is interesting to note that another study demonstrated that SAHA by inducing DR5 expression on the cell surface and downregulating c-IAP-2 and Bcl-xL is able to sensitize resistant melanoma cells to human cytotoxic T-lymphocytes (CTL) [201,202]. Coordinating the up-regulation of death receptors and downregulation of cFLIP and Bcl-2 family members by HDACi (Figure 11a) has also been described to a variable extent in leukemia ([66,118,120,121,203,204], breast carcinoma [199,205], lung carcinoma [119,178,199], glioblastoma [197,206], neuroblastoma [168,207], hepatocellular carcinoma [208,209], or bladder carcinoma [119,210,211]. Regardless of the molecular mechanism, HDACi are often able to restore TRAIL-induced apoptosis in resistant cancer cells [95,125,135,137,178,185,197,200,201,202,206,207,212,213,214,215,216,217,218,219].

By inhibiting histone deacetylases, HDACi induce the accumulation of acetylated lysine residues on histones, which leads to the relaxation of the chromatin and allows gene expression (Figure 11b). However, gene expression or repression is also regulated by BET (bromodomain and extra terminal) proteins, which have a strong affinity for acetylated histones. BET proteins represent thus an attractive target for cancer therapy, in particular when combined with HDACi, since it was found that BET inhibitors such as JQ1 can inhibit c-Myc [220]. Combination treatments associating HDACi and BETi induce strong apoptosis in advanced-stage cutaneous T-cell Lymphoma through inhibition of c-Myc and upregulation of pro-apoptotic proteins, including DR4, DR5 and TRAIL [221]. Interestingly, BETi, such as JQ1, OTX015, CPI-0610 or I-BET762, in non-small cell lung carcinoma (NSCLC), induce strong and preferential inhibition of XIAP and c-FLIP [222], suggesting that like HDACi, these inhibitors are able to coordinate selectively and differentially the expression levels of pro-apoptotic and anti-apoptotic proteins regulating TRAIL-induced cell death. Alone, JQ1 was found to inhibit both the short and the long isoform of c-FLIP in the NSCLC H157, H1299 and A549 cell lines and combined with TRAIL, synergistically induced apoptosis [223]. Its effect on c-FLIP was proposed to occur regardless of c-Myc, through a mechanism involving the proteosomal degradation [223].

This caspase-8 inhibitor, prevents apoptosis induced by death-domain containing receptors of the TNF superfamily [156,224], as well cell death induced by TLR3 [225]. c-FLIP is targeted by HDAC in almost all tumor cell type [66,69,121,205,208,209,226,227,228,229,230,231]. Mechanistically, this short-lived protein was found to interact, in colorectal cancer cells, with the DNA repair protein Ku70 [232]. By inducing Ku70 acetylation, vorinostat was found in these cells to disrupt Ku70 binding to c-FLIP, allowing polyubiquitination and degradation of c-FLIP by the proteasome (Figure 11c). These effects were recapitulated by an HDAC6 inhibitor [232]. It was next found in the hepatocellular carcinoma cell line HepG2 that SIRT1 interacts with c-FLIP and Ku70, inhibiting Ku70 acetylation and leading to c-FLIP stabilization [69]. Inhibiting SIRT1 expression using siRNAs or its function using the SIRT1 inhibitor amurensin G, a specific SIRT1 inhibitor, induced c-FLIP degradation and increased cell death-induced by TRAIL [69]. It should be noted here, though, that SIRT1 was also found to inhibit c-Myc, repressing in turn expression of ATF4, CHOP and DR5 ([69] and Figure 11c). Consistent with these findings, it has been demonstrated that DR5 and DR4 expression levels increase in cells submitted to acute or persistent endoplasmic reticulum (ER) stress through the regulation of ATF4 and CHOP, and that the upregulation of DR4 an DR5, in these conditions, can contribute to apoptosis induced by ER stress in a TRAIL-independent manner [161,162,163]. Notably, salermide, a class III HDAC inhibitor, was found to induce an ER stress leading to the upregulation of DR5 at the cell surface of NSCL cells and to trigger apoptosis, mediated at least in part by DR5 [233]. Salermide-induced DR5 upregulation was mediated through the activation of the ER-stress sensors inositol-requiring enzyme 1 α (IRE-1α), binding immunoglobulin protein (Bip), cyclic AMP-dependent transcription factor (ATF3), ATF4, and CCAAT/-enhancer-binding protein homologous protein (CHOP) [233].

Finally, histone deacetylase inhibitors have been found to act at the very proximal level of the TRAIL signal transduction pathway. For instance, in chronic lymphocytic leukemia cells, romidepsin has been described to facilitate FADD recruitment to DR4, enhancing thus TRAIL DISC formation and activation [203]. In leukemia cells, HDAC inhibitors, including VPA, SAHA, TSA and entinostat, used alone, have also been demonstrated to induce apoptosis through upregulation of TRAIL itself contributing to their pro-apoptotic activity ([135,137] and Figure 9). Along the line, it may be worth mentioning that association of the protein synthesis inhibitor homoharringtonine, also known as omacetaxine mepesuccinate, recently approved by the FDA, acts synergistically with vorinostat to induce apoptosis in a TRAIL/TRAIL receptor-dependent manner in acute myeloid leukemia cells [234]. Altogether these findings further highlight the importance of the epigenetic regulation of TRAIL and its cognate receptors in cancer cells. 

## 8. Conclusions

The epigenetic modification of histone and non-histone proteins plays a central role in the physiopathology of cancer as a result of imbalanced acetylation/deacetylation and methylation/demethylation ratios. While several HDACi are approved by the FDA as a single agent in the treatment of different type of cancers, combination of HDACi with chemotherapeutic agents, epigenetic regulators such as demethylating agents or Bet inhibitors may increase their therapeutic potential. Alternatively, since HDAC inhibitors like TRAIL display apparent selective antitumoral properties, their association with TRAIL, which restores the sensitivity of resistant tumor cells to apoptosis by coordinating the downregulation of antiapoptotic genes, including c-FLIP, XIAP, and survivin or the upregulation of TRAIL agonist receptors (DR4 and DR5), could represent a promising antitumoral approach. To date, only one clinical trial has been conducted to determine the safety and tolerability of combinations associating a TRAIL derivative and a HDACi. Conatumumab, a fully human monoclonal agonist antibody against DR5, was combined with vorinostat in a phase Ib study with patients suffering from low grade lymphoma, mantle lymphoma, diffuse lymphoma, or Hodgkin’s disease. While the overall response of this study was reported to be poor (Clinical study report 20060340) despite the fact that the combination demonstrated a safety profile, the evaluation of TRAIL, or derivatives combined to HDACi or other epigenetic regulators, such as demethylating agents, is likely to represent an interesting therapeutic opportunity to treat patients suffering from cancer.

## Figures and Tables

**Figure 1 cancers-11-00850-f001:**
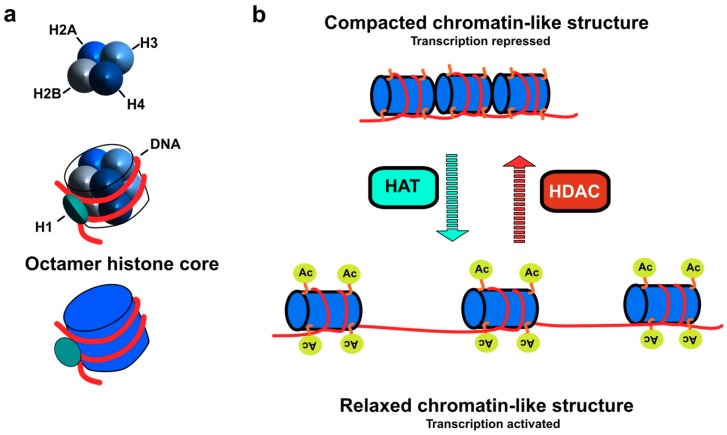
Illustration of histone core composition and impact of chromatin relaxation or compaction by histone acetyltransferases (HAT) or histone deacetylases (HDAC) on gene expression (**a**) Histone protein consists of five main types H1, H2A, H2B, H3, and H4. Two copies of each histone type (2H2A, 2H2B, 2H3, and 2H4) constitute the octamer histone core (nucleosome), wrapped by approx. 146 bp of DNA. H1 is known as linker histone that is associated to linker DNA between each nucleosome. (**b**) Epigenetic modification of histone proteins by HDAC/HAT. HAT stimulates transcriptional activation of the tumor suppressor genes via acetylation of N-ε-lysine residues of histone results in more relaxed chromatin, which facilitates the accessibility of target gene promoter to transcription factors. Conversely, HDAC induces transcriptional repression of tumor suppressor genes by deacetylating N-ε-lysine residues of histone in which the chromatin adopts compacted structure conformation, thus hiding the target gene promoter from the transcription factors.

**Figure 2 cancers-11-00850-f002:**
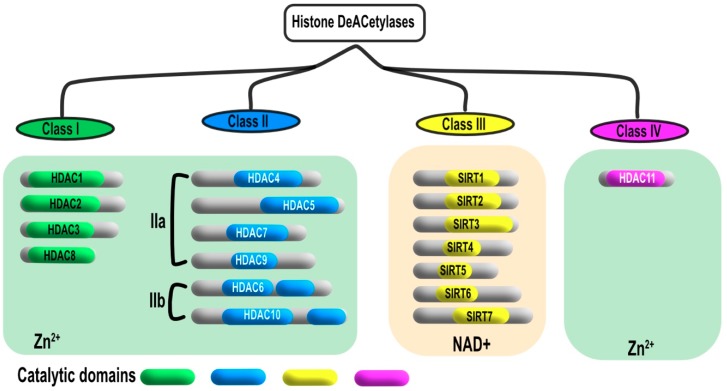
Different classes of HDACs. Four different classes of HDACs were identified in mammalian cell based on their sequence homology and structure similarity to the corresponding yeast HDACs. Class I is homologous to the yeast reduced potassium dependency 3 (RPD3) and includes HDAC1, 2, 3, 8. Class II is homologous to yeast Hda1 and divided into two sub classes, class IIa HDAC4, 5, 7, 9 and class IIb HDAC6, 10. Class III is also known as sirtuins that contain SIRT1-7. Class IV has its own character and shows structural similarity to classes I and II.

**Figure 3 cancers-11-00850-f003:**
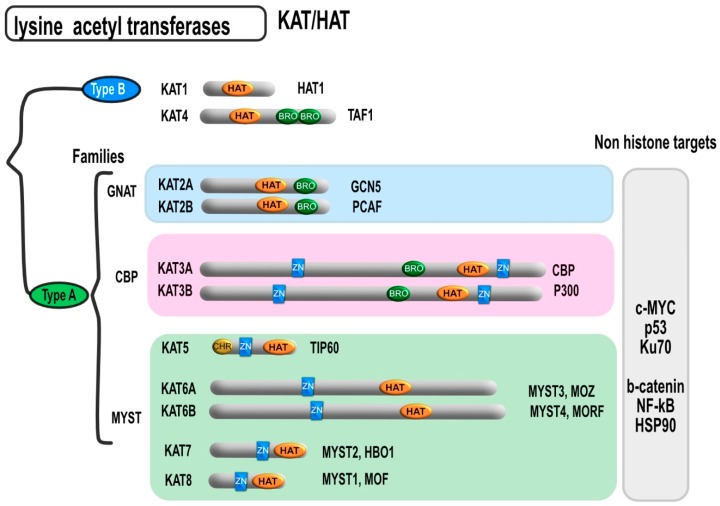
Different HAT/KAT subtypes and their domains. Besides the lysine acetyltransferase domain, KAT can harbor a bromodomain that can facilitate interactions with modified histones. The MYST and CBP families also contain a zinc-binding domain which facilitates DNA binding. Some MYST family HATs contain in addition an N-terminal chromodomain, which binds to methylated lysine residues. HAT = catalytic histone acetyltransferase domain, BRO = bromodomain, CHR = chromodomain, ZN = zinc-binding domain.

**Figure 4 cancers-11-00850-f004:**
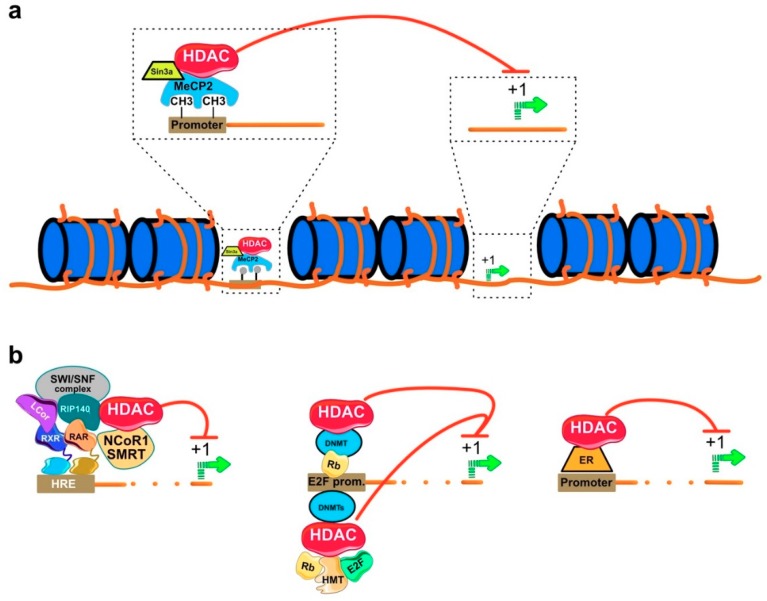
Illustrating the physiological function of HDAC. HDAC can be recruited to gene promoter associated with coregulatory proteins including (**a**) methyl binding protein (MeCP2), (**b**) DNA methyl transferases (DMNT), Estrogen receptor (ER), and transcription factors Rb and E2F, which are responsible for gene silencing and subsequent chromatin remodeling [37].

**Figure 5 cancers-11-00850-f005:**
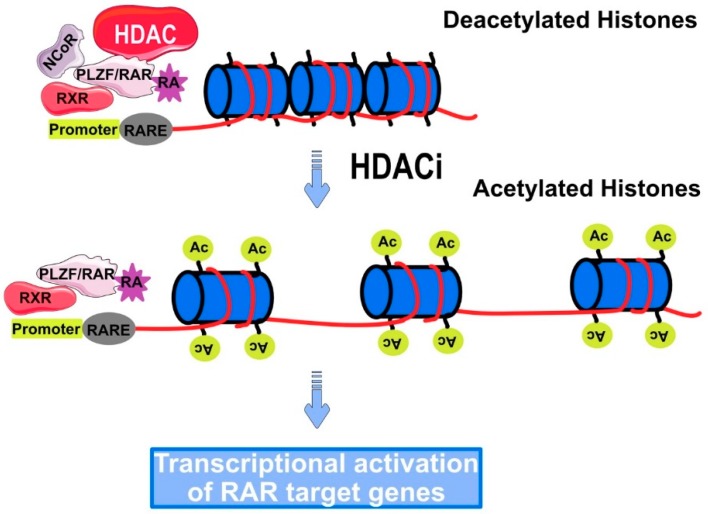
Recruitment of HDAC to retinoic acid-responsive elements (RARE) in Promyelocytic leukemia (PML). HDAC forms a multiprotein complex with RAR-PLZF, resulting in the transcriptional repression of retinoic acid receptor (RAR) target gene, responsible for normal differentiation and proliferation of myeloid cells. HDACi can reverse this case by restoring the sensitivity of PML to retinoic acids.

**Figure 6 cancers-11-00850-f006:**
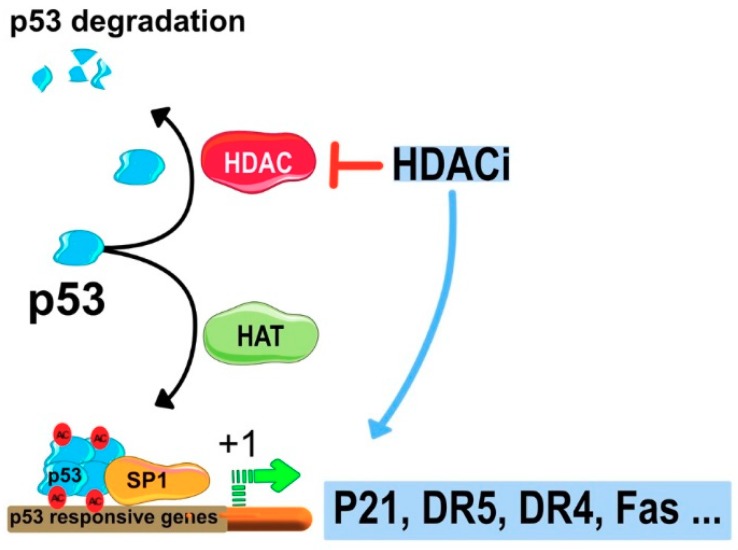
Schematic representation of the molecular mechanism of HDCAIs-induced cell cycle arrest and induction of apoptotic cell death. P21 gene promoter Sp1 can bind HDAC multi protein complex repressing gene transcription. Inhibition of HDAC activates transcription of p21 that stimulates cell cycle arrest. HDACIs can also induce apoptosis via stimulation of tumor necrosis factor (TNF) protein members like TRAIL and CD95.

**Figure 7 cancers-11-00850-f007:**
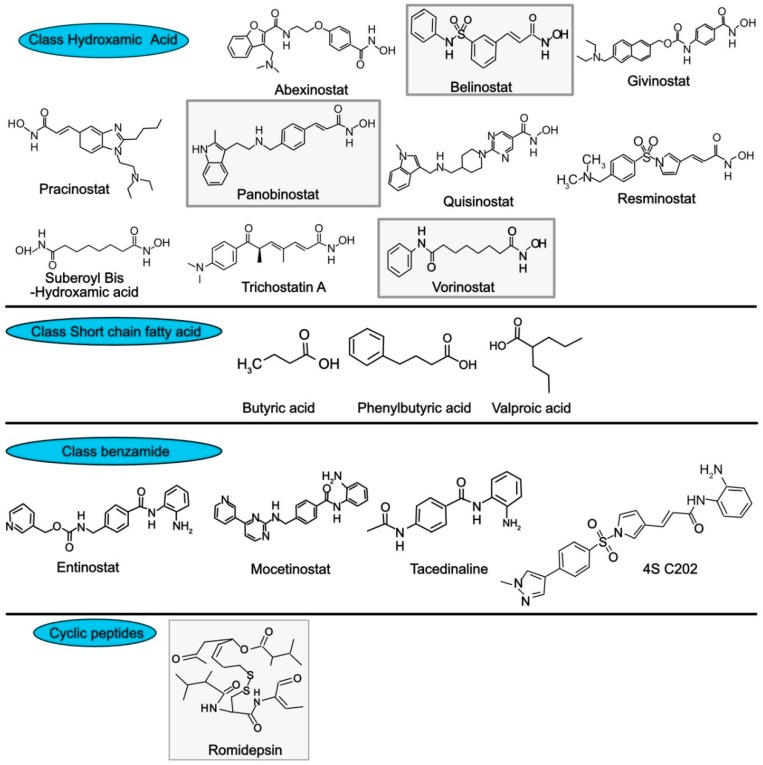
Structure of selected HDAC inhibitors. FDA approved inhibitors are highlighted in the grey boxes.

**Figure 8 cancers-11-00850-f008:**
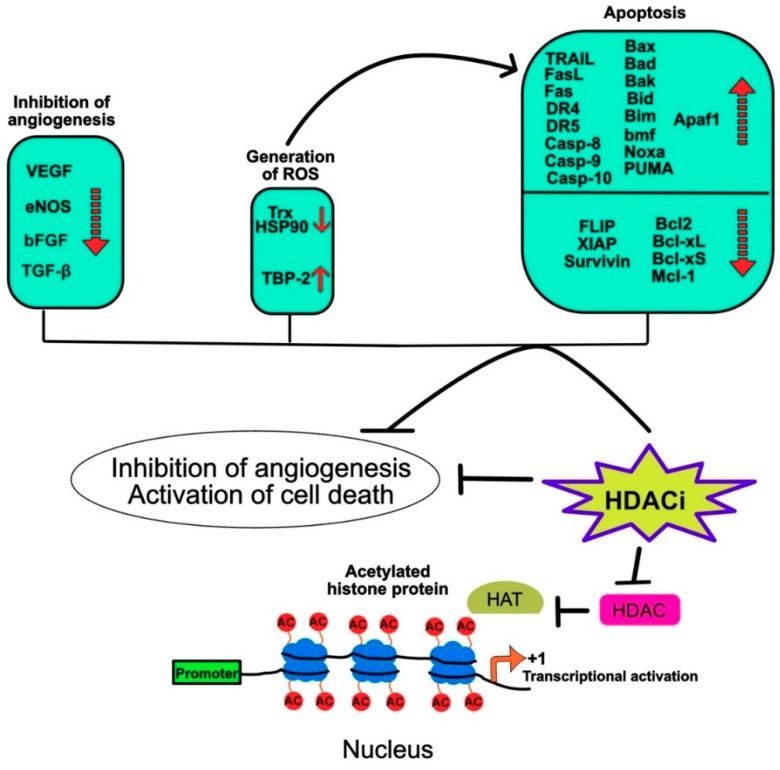
Schematic representation of the main molecular events-induced by HDAC inhibitors preventing tumor metastasis and inducing tumor cell death.

**Figure 9 cancers-11-00850-f009:**
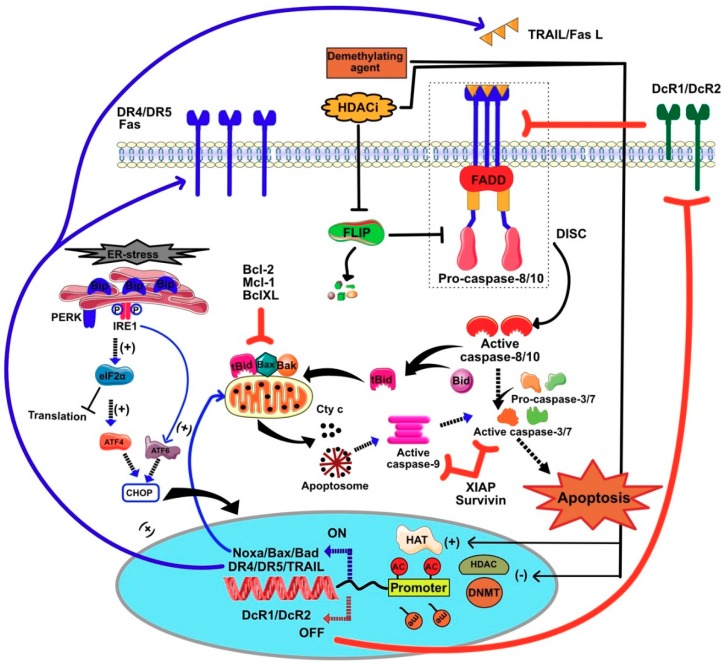
Schematic diagram summarizing the different cell death signaling pathways-induced by epigenetic modification of chromatin following treatment with HDAC inhibitors and demethylating agents.

**Figure 10 cancers-11-00850-f010:**
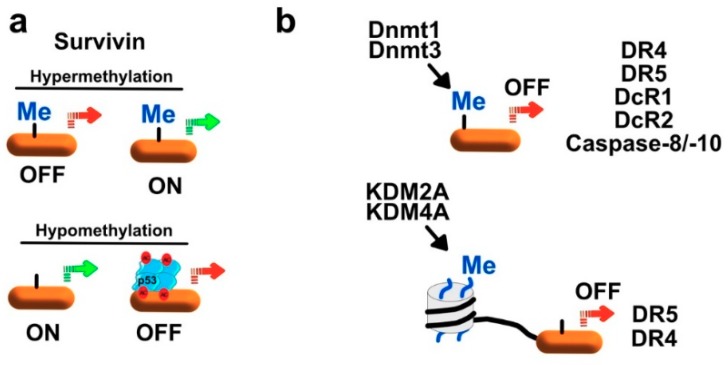
(**a**) Epigenetic regulation of survivin in transformed cells. The promoter region of survivin is hypermehylated thus inhibiting TRAIL-dependent apoptotic signaling pathway. In contrast, hypomethylation of survivin promoter facilitates the binding of p53 repressing its transcriptional activation. (**b**) Epigenetic alterations represented in the methylation of TRAIL death receptors (DR4, DR5, DcR1, and DcR2) confer significant contributions to TRAIL-resistance in different type of tumors. Several DNA (Dmnt1&Dmnt3) and histone (KDM2B&KDM4A) methylases control the regulation of these receptors on the genomic level. Demethylation drugs such as azacitidine and decitabine as well as HDACIs like SAHA, TSA, and sodium butyrate can restore TRAIL sensitization.

**Figure 11 cancers-11-00850-f011:**
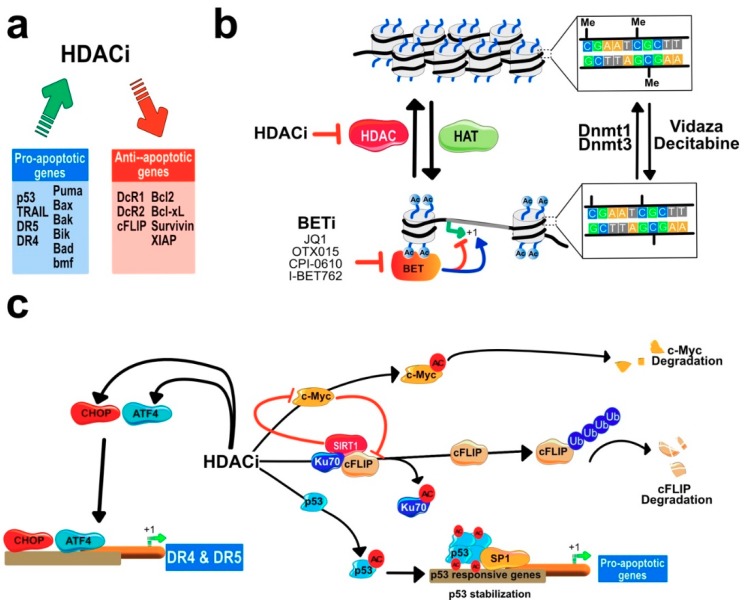
Schematic representation of epigenetic events regulating TRAIL-induced apoptosis. (**a**) Main genes regulated by HDACi, either positively or negatively. (**b**) Illustration of the main epigenetic events affecting chromatin compaction and inhibitors that stimulate chromatin relaxation-based histone acetylation. Bromo-and extra terminal domains (BET) specifically bind to acetylated histones, regulating either positively or negatively genes in the neighborhood. Methylation of CpG island by DNA methylases also contribute to gene silencing. (**c**) HDACi can also regulate gene expression and increase sensitivity to TRAIL-induced apoptosis through acetylation of non-histone proteins. Acetylation of c-Myc induce its degradation whereas acetylation of Ku70 and p53, on the other hand, lead to enhanced TRAIL-induced cell death due to degradation of c-FLIP and upregulation of pro-apoptotic gene expression. HDACi can also induce regulation of TRAIL death receptors (DR4 & DR5) gene expression through the ER-stress sensors ATF4 and CHOP, see text.

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
