# Peer review of "Epigenetic Regulation of TRAIL Signaling: Implication for Cancer Therapy"

_cancers, 2019, doi:10.3390/cancers11060850_

Round 1
Reviewer 1 Report
The manuscript by M.I.Y Elmallah and O. Micheau comprehensively reviewed the biology of histone acetylation and its regulatory mechanism in cancer, followed by a discussion of epigenetic control of TRAIL signaling, a potential target for cancer therapy. This is a well written manuscript with information helpful for both researchers and clinicians working in the field of TRAIL-based cancer therapy. My comments are as follows.
Specific Comments:
1. The title “Epigenetic regulation of TRAIL signaling” does not thoroughly reflect what is included in the content of the manuscript and should be more cancer specific.
2. A separate section about DNA methylation and its regulation in cancer should be added to sections 1~6 before introducing “epigenetic regulation of TRAIL proapoptotic signaling” (page 13), since DNA methylation constitutes an important part of epigenetic control of gene expression in cancer and therefore cannot be ignored.
Author Response
Dear Referee, let us first of all thank you for your comments. Revision are highlighted in yellow.
We have changed the title as requested and added a paragraph on methylation.
Reviewer 2 Report
The manuscript entitled “Epigenetic regulation of TRAIL signaling examines the role of HDACs in cancer etiology and recent advances in the development of HDACi as potential drugs synergizing with TRAIL's pro-apoptotic potential. The review is well compiled and illustrations provided are generally of good quality. However, the authors need to consider few important points as listed below before it can be acceptable for publication.
1. The advantage of HDACs as a druggable target in comparison to other existing epigenetic markers should be discussed.
2. The role of HDACs in regulating various important hallmarks of cancer should be discussed.
3. Conclusion section is short and needs elaboration.
4. The authors should provide their own justification and relevance of the study. This will help the readers to understand the importance of the paper.
5. A table showing the various ongoing/completed clinical trials with HDACi should be included.
Author Response
Dear Referee, let us first of all thank you for your comments. Revision are highlighted in yellow.
We would like to emphasize that our review is not intended to be an exhaustive presentation or update of epigenetic regulation and cancer, there are dozens of them in Pubmed, but a review dedicated to TRAIL signal transduction regulation by epigenetic regulators, emphasizing how the latter maybe profitable to cancer therapy.
It is our belief that comments on advantage of HDACi vs other epigenetic markers and cancer hallmark go largely beyond the scope of our review and need not to be addressed specifically since most of these points are discussed in our paper. Besides, adding the list of current HDACi clinical trials, per se, would not had much to our review. Readers can find such table in other reviews.